# Family caregivers' emotional and communication needs in Canadian pediatric emergency departments

Samina Ali[1,2&]*, Claudia Maki[1&], Asa Rahimi[1], Keon Ma[3], Maryna Yaskina[2], Helen Wong[4], Antonia Stang[3], Tania Principi[3], Naveen Poonai[5], Serge Gouin[6], Sylvia Froese R. N.[7], Paul Clerc[8], Redjana Carciumaru[9], Waleed Alqurashi[10], Manasi Rajagopal[1], Elise Kammerer[1], Julie Leung[11], Bruce Wright[1,2], Shannon D. Scott[12], on behalf of the Pediatric Emergency Research Canada Family Needs Study Group[¶]

1 Department of Pediatrics, Faculty of Medicine & Dentistry, University of Alberta, Edmonton, Alberta, Canada, 2 Women & Children's Health Research Institute (WCHRI), Edmonton, Alberta, Canada, 3 Department of Pediatrics, Cumming School of Medicine, University of Calgary, Calgary, Alberta, Canada, 4 Faculty of Health, Dalhousie University, Halifax, Nova Scotia, Canada, 5 Department of Emergency Medicine, Section of Pediatric Emergency Medicine, Schulich School of Medicine & Dentistry, Western University, London, Ontario, Canada, 6 CHU Ste Justine, Montreal, Quebec, Canada, 7 Children's Hospital Research Institute of Manitoba, University of Manitoba, Winnipeg, Manitoba, Canada, 8 Department of Emergency Medicine, University of British Columbia, Vancouver, British Columbia, Canada, 9 Department of Pediatrics, McMaster University, Hamilton, Ontario, Canada, 10 Department of Pediatrics, University of Ottawa, Ottawa, Ontario, Canada, 11 Community Engagement Stakeholder, Edmonton, Alberta, Canada, 12 Faculty of Nursing, University of Alberta, Edmonton, Alberta, Canada

& These authors contributed equally to this work.
¶ The complete membership of the author group can be found in the Acknowledgments.
* sali@ualberta.ca

**Data Availability Statement:** Data cannot be shared publicly because of consent and confidentiality reasons. Data are available from the PEAK Research Team via Patricia Candelaria

## Abstract

### Objectives

To describe the extent to which caregivers' emotional and communication needs were met during pediatric emergency department (PED) visits. Secondary objectives included describing the association of caregiver emotional needs, satisfaction with care, and comfort in caring for their child's illness at the time of discharge with demographic characteristics, caregiver experiences, and ED visit details.

### Study design

Electronic surveys with medical record review were deployed at ten Canadian PEDs from October 2018 –March 2020. A convenience sample of families with children <18 years presenting to a PED were enrolled, for one week every three months, for one year per site. Caregivers completed one in-PED survey and a follow-up survey, up to seven days post-visit.

### Results

This study recruited 2005 caregivers who self-identified as mothers (74.3%, 1462/1969); mean age was 37.8 years (SD 7.7). 71.7% (1081/1507) of caregivers felt their emotional needs were met. 86.4% (1293/1496) identified communication with the doctor as good/very

(stacruz@ualberta.ca) or the corresponding author Dr. Samina Ali (sali@ualberta.ca) for researchers who meet the criteria for access to confidential data.

**Funding:** This work was supported by the Women and Children's Health Research Institute via both a Clinical/Community Research Integration Support Program grant secured by Drs. Samina Ali and Shannon Scott (2018-2022) and a trainee grant secured by Dr. Claudia Maki (2022-2023) through a generous donation from the Stollery Children's Hospital Foundation. The funders of the study had no role in study design, data collection, data analysis, data interpretation, or writing of the report. There was no additional external funding received for this study.

**Competing interests:** The authors have declared that no competing interests exist.

good and 83.4% (1249/1498) with their child's nurse. Caregiver involvement in their child's care was reported as good/very good 85.6% (1271/1485) of the time. 81.8% (1074/1313) of caregivers felt comfortable in caring for their child at home at the time of discharge. Lower caregiver anxiety scores, caregiver involvement in their child's care, satisfactory updates, and having questions adequately addressed positively impacted caregiver emotional needs and increased caregiver comfort in caring for their child's illness at home.

## Conclusion

Approximately 30% of caregivers presenting to PEDs have unmet emotional needs, over 15% had unmet communication needs, and 15% felt inadequately involved in their child's care. Family caregiver involvement in care and good communication from PED staff are key elements in improving overall patient experience and satisfaction.

## Introduction

Pediatric emergency departments (EDs) have been identified as loud, chaotic, unfamiliar and inherently stressful for families, and often lacking in longitudinal provider-patient relationships. While family presence and the provision of point-of-care psychotherapeutic support is now an accepted practice for high acuity resuscitation situations [1–3], these interventions do not address the individual psychosocial needs of families in less acute emergency care situations [1]. Our research team previously identified caregiver stress as a variable associated with early returns to pediatric Eds [4]. Still, the source of this increased caregiver stress has not been well explored and may suggest potential unmet emotional needs of families in the ED setting [4].

Emotional needs of families presenting to the pediatric ED with specific conditions have been previously described in small, single center studies, and share some common themes regarding caregiver needs [5–8]. For infants admitted to the ED with fever, caregivers describe emotions such as worry, fear, and stress, related to fear of the unknown, stress during procedures, and the emotional distress of hospital admission impact on the entire family [6]. Minor head trauma creates anxiety for caregivers, specifically around the worry that their child has suffered brain damage [5]. Caregivers of children presenting to hospital with mental health crises endorse feelings of stress, especially fear of inpatient admission [7]. Similarly, caregivers of children with sickle cell disease have been found to experience emotional distress that is directly correlated with their child's pain [8].

Information, communication, shared decision making, trust and connection with the health care team have all been identified as key elements in supporting the emotional needs of caregivers in the pediatric intensive care setting, particularly when they have to make difficult decisions regarding the care of their child [9]. In the general ED setting, only one national study of caregivers reports their comfort level with their strategies to deal with their own anxiety related to the ED visit and ability to comfort their child needing ED care as 7/10, leaving significant room for improvement [10]. To date, there remains a paucity of research addressing the global emotional needs of caregivers during ED visits.

Patient and family-centered care (PFCC) acknowledges that emotional support is an integral component to health care, leads to better health outcomes and satisfaction, and enhances caregivers' confidence in their caregiving roles [11]. Patient and caregiver satisfaction are intimately related, with caregiver satisfaction being dependent on the quality of interpersonal interaction, communication, empathy, and compassion demonstrated, and influencing the child's experience in the pediatric setting [12, 13]. Better understanding the emotional and

communication needs of caregivers in the ED could identify opportunities to be more responsive to families' needs. To date, no studies have directly assessed the global emotional and communication needs of families in the pediatric ED setting.

Our study objectives were to describe the extent to which caregivers' emotional and communication needs were met during pediatric emergency department (ED) visits. Secondary objectives included describing the association of caregiver emotional needs, satisfaction with care, and comfort in caring for their child's illness at the time of discharge with demographic characteristics caregiver experiences, and ED visit details.

## Materials and methods

### Study design and setting

This manuscript presents a sub-study of a descriptive cross-sectional survey with medical record review [14]. Ten of 15 Canadian pediatric EDs participated in enrolling a convenience sample of families. The Stollery Children's Hospital (Edmonton, AB) was the lead site. Other participating sites included: Alberta Children's Hospital (Calgary, AB), Children's Hospital at London Health Sciences Center (London, ON), Children's Hospital of Eastern Ontario (Ottawa, ON), McMaster Children's Hospital (Hamilton, ON), BC Children's Hospital (Vancouver, BC), Winnipeg Children's Hospital (Winnipeg, MB), The Hospital for Sick Children (Toronto, ON), CHU Sainte Justine (Montreal, QC), IWK Health Centre (Halifax, NS). The annual census for the involved institutions (45,000–80,000 per annum). All participating EDs were academic tertiary care hospitals and members of Pediatric Emergency Research Canada [15]. Caregivers with lived experience were co-investigators and contributed to the study from its inception including study design, results interpretation, and knowledge translation plan.

A sample of approximately 2000 caregivers presenting to the pediatric ED were targeted to be enrolled in the study based on a goal of 200 caregivers recruited per site. Enrollment occurred over a one-week period per season, over 12 months, for a total of 4 weeks of recruitment per site. This recruitment method, used in prior pediatric ED research, allowed for capturing seasonal variation in family needs, overcrowding, and disease presentation [16, 17]. Caregivers were recruited from October 2018—March 2020. Ethics approvals were obtained from the Research Ethics Boards (REB) of all sites, including the University of Alberta Health REB; the University of Calgary Conjoint Health REB; the University of British Columbia Children's and Women's REB; the Western University Health Sciences REB; the Children's Hospital of Eastern Ontario REB; the CHU Ste Justine REB; the SickKids REB; the IWK Health Centre REB; the McMaster University REB; and the University of Manitoba Health REB. The site leads and research coordinators had access to information that could identify individual participants at their own site, only, during data collection. All data were anonymized prior to analyses and sharing of results within the research team.

### Selection of participants

Included in this study were consenting families with children aged 0–17 years presenting to a participating pediatric ED with any chief complaint. A prerequisite for participation in this study was caregiver ability to read and write English or French. Families were excluded if (a) the child was medically unstable throughout their ED stay, (b) there was suspicion of child abuse, (c) the child was presenting with an altered level of consciousness, or (d) if the accompanying person was not a legal guardian. Each participant could take part in the study only once.

## Data collection

Research staff were trained on the study by the principal investigator and national coordinator. The research staff then conducted eligibility screening and performed data entry directly into a Research Electronic Data Capture (REDCap) database [18], a secure, online data entry system hosted at the University of Alberta. Research staff obtained written informed consent from participating caregivers. Following published recommendations [19], two novel survey tools were developed to assess caregiver needs and experiences. The surveys underwent item generation and reduction (6-member expert panel), as well as pre-testing (8 participants contacted via email), pilot testing (10 participants in ED setting), and sensibility testing (10 participants in ED setting). The final surveys included an assessment of emotional needs and satisfaction during their visit as well communication needs. Caregivers first completed a brief electronic survey in the ED (5–10 minutes), and a follow up survey via email or phone (10–15 minutes), up to seven days after their visit. Participants could skip any questions that they wished; missed responses were not permuted. A medical record review was conducted to obtain additional ED visit details. A 5$ gift card was provided to each family upon completion of the surveys.

## Measurements

After enrollment, trained research staff administered electronic surveys to caregivers in the ED to collect brief demographic information, State-Trait Anxiety Inventory (STAI) [20] assessment and caregiver health literacy score utilizing the Newest Vital Sign (NVS) [21] instrument. The STAI is a validated 20 item tool assessing adult state (i.e. experience-related) anxiety; possible scores range from 20–80, with higher scores indicative of increased anxiety [20]. The NVS is a brief, validated tool for measuring health literacy [21], that has been previously employed in the ED setting, with scores ranging from 0 to 6 [22]. Caregiver scores are grouped into three categories: adequate literacy (4–6), possibility of limited literacy (2–3), or high likelihood of limited literacy (0–1) [21]. In the follow-up survey, caregivers were asked to rate how well their emotional and communication needs were met using a 5-point Likert Scale (1 = very poor/little, 5 = very well/much). When required for analyses, Likert scaled questions were dichotomized into 'needs not met' (Likert 1, 2 or 3) and 'needs met' (Likert 4 or 5).

## Outcomes

The primary outcome was a quantitative description of the extent to which caregivers reported emotional and communication needs were met during their pediatric ED visit (as measured via 5-point Likert scale). Secondary outcomes included describing the association of emotional needs, satisfaction with care, and caregiver comfort in caring for their child's illness at the time of discharge, with demographic characteristics, caregiver experiences, and ED visit details.

## Analysis

Descriptive statistics (means, medians, standard deviations, interquartile ranges) were completed for continuous variables (e.g., LOS, age), while frequency distributions summarized categorical variables (e.g., sex, health literacy). Emotional and communication needs and satisfaction were measured with a 5-point Likert scale with 1 meaning negative ("Not at all") and 5 meaning positive ("Very much")' these were summarized as frequency distributions. For the regression analyses, outcomes (e.g., emotional and communication needs, caregiver comfort in caring for their child's illness at time of discharge) were recategorized into binary variables. Answers 1, 2, and 3 were considered as "No" (emotional or communicational needs not met or caregiver is not comfortable caring for their child) while answers 4 and 5 were considered as

"Yes" (emotional or communicational needs are met or caregiver is comfortable caring for their child). Multivariable logistic regression was used for the outcomes to ascertain effects of specific independent variables (e.g., triage category, health literacy, state anxiety, caregiver age, child age, number of other children) on caregivers' reported emotional and communication needs and their satisfaction. Results from the regression models were reported using odds ratios and 95% confidence intervals. A p-value less than 0.05 was considered statistically significant. All statistical analyses were performed using SAS Version 9.4 (SAS Institute Inc., Cary, NC, USA).

## Results and discussion

### Demographic characteristics

A total of 2005 eligible caregivers were enrolled during the study period. Most caregivers were mothers (74.3%, 1462/1969) followed by fathers (24.2%, 476/1969) and grandparents (0.7%, 13/1969). Mean caregiver age was 37.8 years (SD 7.7), and 23.9% (453/1899) of caregivers had only one child. The mean age for the child attending the ED was 5.9 years (SD 5.1), and 51.9% (1040/2003) were male. Mean caregiver health literacy score was 4.3 (SD 1.9). The proportion of caregivers with adequate health literacy was 71.5% (1399/1957), possibility of limited literacy 16.5% (323/1957) and high likelihood of limited literacy 12% (235/1957). Mean STAI score for caregivers was 37.9 (SD 11.1). (Table 1 and S1 Table).

### Caregiver emotional needs

The proportion of caregivers who reported emotional needs being met during their ED visit were 71.7% (1081/1507). Staff physicians (35.6% (535/1502)), bedside nurses (24.2% (364/1502)), residents/medical students (13.8% (207/1502)) and triage nurses (12.4% (186/1502)) were identified as providing the best emotional support to caregivers. The proportion of caregivers that felt their child's privacy was respected was 90.2% (1361/1508). Only 16.8% (253/1505) of caregivers wondered if they should have come to the hospital sooner. The proportion of caregivers who reported feeling scared/very scared was 11.3% (171/1507), while 55.4% (835/1507) caregivers reported not feeling scared ('very little'); overall, 55.5% (410/739) reported that the ED health care team (i.e., bedside nurse, physician, child life specialist, social worker, etc) made them feel better about feeling scared. The proportion of caregivers who felt being treated differently or poorly based on race, religion, sexual orientation, language or disability were 2.4% (36/1512). Of note, 4.9% (74/1503) of caregivers reported seeing a traumatic or upsetting situation for another child while in the ED. (Table 2).

### Caregiver communication needs

A total of 83.4% (1249/1498) of caregivers identified communication needs with their child's nurse as met, while 86.4% (1293/1496) identified communication needs with their child's doctor as met. Involving caregivers in their child's care was reported as good/very good 85.6% (1271/1485) of the time. Information provided to caregivers regarding their child's condition was reported as good/very good 81.5% (1218/1494) of the time. Satisfactory updates to the caregiver while the child was in ED was reported as good/very good in 72.6% (1081/1488) of cases; questions and concerns were answered well/very well for 83.2% (1239/1489). Caregivers felt comfortable caring for their child's injury/illness at home 81.8% (1074/1313) of the time. (Table 3).

### Factors associated with meeting caregiver needs

Relevant univariate modeling is presented in S2 Table. In multivariable logistic regression modeling, the following factors were associated with caregiver emotional needs being met:

**Table 1. Characteristics of caregivers and children.**

| Characteristics of caregivers | All caregivers n(%) |
|---|---|
| **Caregiver age, years (n = 1913)** | |
| Mean (SD) | 37.8 (7.7) |
| **Relationship to the child being seen in ED (n = 1969)** | |
| Mother | 1462 (74.3) |
| Father | 476 (24.2) |
| Grandparent | 13 (0.7) |
| Other | 18 (0.9) |
| **Number of total children (n = 1899)** | |
| 1 | 453 (23.9) |
| 2 | 851 (44.8) |
| 3 | 375 (19.7) |
| >3 | 220 (11.7) |
| **Main spoken language at home (n = 1964)** | |
| English | 1425 (72.6) |
| French | 206 (10.5) |
| Chinese (including Mandarin, Cantonese) | 46 (2.3) |
| Spanish | 44 (2.2) |
| Hindi/Urdu/Punjabi | 41 (2.1) |
| Arabic | 40 (2.0) |
| Other | 162 (8.4) |
| **Highest level of education (n = 1922)** | |
| University/Professional Degree | 997 (51.9) |
| Diploma/Certificate | 406 (21.1) |
| Some Post-secondary/University | 328 (17.1) |
| Some High School/High School | 184 (9.6) |
| Elementary School | 7 (0.4) |
| **Annual household income (n = 1706)** | |
| $\leq$ $25,000 | 135 (7.9) |
| $25,001 to $50,000 | 267 (15.7) |
| $50,001 to $75,000 | 239 (14.0) |
| $75,000 to $100,000 | 298 (17.5) |
| > $100,000 | 767 (45.0) |
| **Caregiver State Trait Anxiety Inventory Score (n = 1816)** | |
| Mean (SD) | 37.9 (11.1) |
| **Caregiver Health Literacy (n = 1957)** | |
| High likelihood of limited literacy (0–1) | 235 (12) |
| Possibility of limited literacy (2–3) | 323 (16.5) |
| Adequate literacy (4–6) | 1399 (71.5) |
| **Characteristics of children** | **All children n(%)** |
| Age, years (n = 2003) Mean (SD) | 5.9 (5.1) |
| Child sex, male | 1040 (51.9) |
| **Previous ED visits (n = 1966)** | |
| 0 | 363 (18.5) |
| 1–5 | 1153 (58.6) |
| 6–10 | 272 (13.8) |
| > 10 | 178 (9.1) |
| **CTAS Score (n = 1991)** | |

(*Continued*)

**Table 1.** (Continued)

| Characteristics of caregivers | All caregivers n(%) |
|---|---|
| 1 –Resuscitation | 9 (0.5) |
| 2 –Emergent | 366 (18.4) |
| 3 –Urgent | 1010 (50.7) |
| 4 –Semi urgent | 528 (26.5) |
| 5 –Non urgent | 78 (3.9) |
| **Discharge Disposition (n = 1993)** | |
| Discharged | 1719 (86.3) |
| Admitted | 257 (12.9) |
| Transferred | 10 (0.5) |
| Other | 7 (0.4) |

caregivers feeling that their child's privacy was respected (1.38 [1.15, 1.65]), ED staff involving the caregiver in their child's medical care (1.60 [1.35, 1.89]), providing satisfactory updates to the caregiver about their child's care (1.33 [1.12, 1.58]), adequately answering caregivers' questions and concerns (1.66 [1.35, 2.04]), and higher caregiver health literacy score (1.09 [1.00, 1.18] for every 1-point increase in score). Factors associated with caregivers' emotional needs being unmet were lower acuity triage scores (0.76 [0.63, 0.92]), being the mother (vs father) (0.64 [0.45, 0.91]), and higher STAI (0.97 [0.95, 0.98] for every 1-point increase in score). Caregivers wondering whether they should have come to the ED sooner, feelings of being scared during the ED visit, and number of other children did not affect the likelihood of caregivers' emotional needs being met. (Table 4).

## Caregiver comfort in caring for child's illness at time of discharge modeling

Relevant univariate modeling is presented in S3 Table. In multivariable logistic regression modeling, factors associated with increased caregiver comfort in caring for their child's illness at the time of discharge included: ED staff involving the caregiver in their child's medical care (1.49 [1.23, 1.81)], providing satisfactory updates about their child's care (1.27 [1.03, 1.55]), and adequately answering caregiver questions and concerns (1.73 [1.38, 2.18]). Factors associated with decreased caregiver comfort in caring for their child's illness at the time of discharge were feeling scared during the ED visit (0.74 [0.63, 0.86]) and higher STAI score (0.97 [0.95, 0.98] for every 1-point increase in score). Caregivers wondering whether they should have come to the ED sooner, relationship to the child, NVS score, and the number of other children did not affect caregiver comfort in caring for their child's illness at the time of discharge. (Table 4).

## Discussion

Our study, which surveyed over 2000 caregivers from two-thirds of Canada's pediatric hospitals, examined the extent to which emotional and communication needs of caregivers are met. We demonstrated that almost 30% of caregivers have unmet emotional needs, over 15% have unmet communication needs, 15% did not feel involved in their child's care during their ED visit and nearly 20% of caregivers did not feel comfortable in managing their child's illness at time of discharge. These unmet needs provide insight into both potential short- and long-term considerations for improving caregiver satisfaction and experience in the ED.

**Table 2. Caregiver emotional needs.**

| How well were your emotional needs (e.g., reassurance, comforted if you were upset) met by the ED staff during your visit? (n = 1507) | n (%) |
|---|---|
| 1 (very little) | 83 (5.5) |
| 2 | 92 (6.1) |
| 3 | 251 (16.7) |
| 4 | 498 (33.0) |
| 5 (very much) | 583 (38.7) |
| **Which healthcare provider provided the best emotional support to you? (n = 1502)** | **n (%)** |
| Doctor in charge/Senior doctor | 535 (35.6) |
| Bedside nurse | 364 (24.2) |
| Trainee doctor (medical student/resident) | 207 (13.8) |
| Triage nurse | 186 (12.4) |
| No one | 115 (7.7) |
| Everyone | 30 (2.0) |
| Social worker | 13 (0.9) |
| Technician (Ortho/RT/X-Ray) | 12 (0.8) |
| Child Life | 8 (0.5) |
| Specialist | 7 (0.5) |
| Other (please specify) | 25 (1.7) |
| **How well was your child's privacy respected? (n = 1508)** | **n (%)** |
| 1 (very little) | 17 (1.1) |
| 2 | 32 (2.1) |
| 3 | 98 (6.5) |
| 4 | 334 (22.1) |
| 5 (very much) | 1027 (68.1) |
| **Did you wonder whether you should have come to hospital sooner? (n = 1505)** | **n (%)** |
| 1 (very little) | 728 (48.4) |
| 2 | 257 (17.1) |
| 3 | 267 (17.7) |
| 4 | 145 (9.6) |
| 5 (very much) | 108 (7.2) |
| **If so, how much did the ED team make you feel better about this? (n = 686)** | **n (%)** |
| 1 (very little) | 93 (13.6) |
| 2 | 54 (7.9) |
| 3 | 159 (23.2) |
| 4 | 152 (22.2) |
| 5 (very much) | 228 (33.2) |
| **Did you feel scared during the ED visit? (n = 1507)** | **n (%)** |
| 1 (very little) | 835 (55.4) |
| 2 | 258 (17.1) |
| 3 | 243 (16.1) |
| 4 | 113 (7.5) |
| 5 (very much) | 58 (3.8) |
| **If so, how much did the ED team make you feel better about this? (n = 739)** | **n (%)** |
| 1 (very little) | 99 (13.4) |
| 2 | 71 (9.6) |
| 3 | 159 (21.5) |
| 4 | 198 (26.8) |
| 5 (very much) | 212 (28.7) |

**Table 3. Caregiver communication needs.**

| How was the overall communication between you and your child's nurse? (n = 1498) | n (%) |
|---|---|
| 1 (very poor) | 31 (2.1) |
| 2 | 51 (3.4) |
| 3 | 167 (11.1) |
| 4 | 423 (28.2) |
| 5 (very good) | 826 (55.1) |
| **How was the overall communication between you and your child's doctor? (n = 1496)** | **n (%)** |
| 1 (very poor) | 21 (1.4) |
| 2 | 48 (3.2) |
| 3 | 134 (9.0) |
| 4 | 452 (30.2) |
| 5 (very good) | 841 (56.2) |
| **How much did the doctors, nurses and other providers involve your child in their own care? (n = 1485)** | **n (%)** |
| 1 (very little) | 91 (6.1) |
| 2 | 85 (5.7) |
| 3 | 238 (16.0) |
| 4 | 416 (28.0) |
| 5 (very much) | 655 (44.1) |
| **How much did the doctors, nurses and other providers involve YOU in your child's care? (n = 1492)** | **n (%)** |
| 1 (very little) | 26 (1.7) |
| 2 | 46 (3.1) |
| 3 | 149 (10.0) |
| 4 | 443 (29.7) |
| 5 (very much) | 828 (55.5) |
| **How clear was the information provided to you about your child's condition? (n = 1494)** | **n (%)** |
| 1 (very unclear) | 38 (2.5) |
| 2 | 61 (4.1) |
| 3 | 177 (11.8) |
| 4 | 432 (28.9) |
| 5 (very clear) | 786 (52.6) |
| **How clear was the information provided to you about your child's tests done in the ED? (n = 979)** | **n (%)** |
| 1 (very unclear) | 29 (3.0) |
| 2 | 42 (4.3) |
| 3 | 123 (12.6) |
| 4 | 234 (23.9) |
| 5 (very clear) | 551 (56.3) |

*(Continued)*

**Table 3.** (Continued)

| How clear was the information provided to you about any medicines your child received? (n = 948) | n (%) |
|---|---|
| 1 (very unclear) | 20 (2.1) |
| 2 | 27 (2.8) |
| 3 | 106 (11.2) |
| 4 | 232 (24.5) |
| 5 (very clear) | 563 (59.4) |
| **How satisfactory were the updates to you about your child's care while you were in the ED? (n = 1488)** | **n (%)** |
| 1 (very unsatisfactory) | 73 (4.9) |
| 2 | 101 (6.8) |
| 3 | 233 (15.7) |
| 4 | 490 (32.9) |
| 5 (very satisfactory) | 591 (39.7) |
| **How well did the emergency staff answer your questions and concerns? (n = 1489)** | **n (%)** |
| 1 (very little) | 32 (2.1) |
| 2 | 63 (4.2) |
| 3 | 155 (10.4) |
| 4 | 485 (32.6) |
| 5 (very much) | 754 (50.6) |
| **How satisfied were you with the information given to you before being discharged/admitted? (n = 1481)** | **n (%)** |
| 1 (very unsatisfied) | 55 (3.7) |
| 2 | 64 (4.3) |
| 3 | 203 (13.7) |
| 4 | 480 (32.4) |
| 5 (very satisfied) | 679 (45.8) |
| **How comfortable were you to care for your child's injury/illness at home, after being sent home from the hospital? (n = 1313)** | **n (%)** |
| 1 (very uncomfortable) | 35 (2.7) |
| 2 | 60 (4.6) |
| 3 | 144 (11.0) |
| 4 | 314 (23.9) |
| 5 (very comfortable) | 760 (57.9) |

Understanding families' emotional needs and experiences in the ED is complex, particularly when caring for children, as the experiences reflect both child and caregiver perspectives [23]. Almost 30% of caregivers felt their emotional needs were not met, suggesting that we can do better to meet the needs of the families we serve. Importantly, ED visits are often a

**Table 4. Adjusted odds ratios for likelihood of emotional needs being met and caregiver comfort in caring for their child's illness at time of discharge.**

| Likelihood of emotional needs being met | | |
|---|---|---|
| **Variable** | **Odds Ratio (95% CI)** | **p-value** |
| CTAS (4 categories, continuous)* | 0.76 (0.63, 0.92) | 0.005 |
| Relationship to child (Mother vs Father) | 0.64 (0.45, 0.91) | 0.01 |
| Did you feel that your child's privacy was respected? | 1.38 (1.15, 1.65) | 0.0005 |
| Did the doctors, nurses, and other providers involve YOU in your child's care? | 1.60 (1.35, 1.89) | < 0.0001 |
| How satisfactory were the updates to you about your child's care in the ED? | 1.33 (1.12, 1.58) | 0.001 |
| Did the emergency staff answer your questions and concerns? | 1.66 (1.35, 2.04) | <0.0001 |
| STAI score** | 0.97 (0.95, 0.98) | < 0.0001 |
| NVS score*** | 1.09 (1.00, 1.18) | 0.053 |
| **Caregiver comfort in caring for their child's illness at time of discharge** | | |
| **Variable** | **Odds Ratio (95% CI)** | **p-value** |
| Did you feel scared during the ED visit? | 0.74 (0.63, 0.86) | 0.0001 |
| Did the doctors, nurses, and other providers involve YOU in your child's care? | 1.49 (1.23, 1.81) | < 0.0001 |
| How satisfactory were the updates to you about child's care in the ED? | 1.27 (1.03, 1.55) | 0.023 |
| Did the emergency staff answer your questions and concerns? | 1.73 (1.38, 2.18) | <0.0001 |
| STAI score** | 0.97 (0.95, 0.98) | 0.0002 |

Needs were rated on a 5-point Likert Scale (1 = very poor/little, 5 = very well/much). All Likert scale variables were dichotomized into 'needs not met' (Likert 1, 2, or 3) and 'needs met' (Likert 4 or 5)

* Calculated for every 1 unit increase in CTAS. Of note, CTAS 1 and 2 combined

** Calculated for every 1 unit of increase in STAI score

*** Calculated for every 1 unit of increase in NVS score

caregiver's first experience with the hospital system, influencing their current healthcare trajectory right from the waiting room, as well as having influence over perceptions of subsequent healthcare encounters in the future [12, 24]. Our study demonstrates the importance of all healthcare professionals working together to meet the emotional needs of caregivers in the ED to improve the overall family experience. Physicians, bedside nurses, trainees and triage nurses were identified as providing the best emotional support to caregivers, supporting the literature asserting that improving the ED patient experience requires engagement from both medical and nursing groups [12].

Caregivers were more likely to have their emotional needs met if they felt their child's privacy was respected, had higher health literacy, a higher acuity presentation, or were the child's father. Privacy and caregiver gender have not been previously explored in relation to emotional needs in the ED. However, prior study has demonstrated that families with low health literacy are more likely to overestimate severity of illness, seek care for lower acuity illness, and desire supportive care (including emotional needs) sooner though the ED [25, 26]. Notably, there was overlap in variables that positively impacted emotional needs being met and increased caregiver comfort in caring for their child's illness at time of discharge. These included lower caregiver anxiety scores, involvement of the caregiver in their child's care, satisfactory updates about their child's condition and care, and caregiver questions and concerns being adequately addressed by HCPs. We know that diagnostic uncertainty is associated with higher return visits to the ED for children and their caregivers; [27] while the connection to emotional needs has not been explicitly made prior to our current study, it stands to reason that feeling unsure and unsafe (i.e., unmet emotional needs) likely has a role in this return visit

behaviour. This needs to be further explored through qualitative inquiry to better understand the relationship between emotional needs and comfort caring for a child after an ED visit.

It has been previously demonstrated that achieving optimal patient and caregiver satisfaction in the ED is dependent on the quality of interpersonal interaction, communication, empathy, and compassion demonstrated [12, 13]. Our study found over 15% of caregivers felt their communication with doctors and nurses were very poor/poor, which is unfortunate as communication and ensuring patients/caregivers are informed leads to greater satisfaction, despite prolonged wait times, increased census, and cramped ED spaces [12, 13, 28]. Further, almost half of caregivers felt that the ED staff were not able to make them feel better when they were scared. Physicians, nurses, and trainees all have a role in meeting the emotional needs of families and were all identified as top sources of support for caregivers. By utilizing a team approach to improve communication (e.g., attending physician asking resident to update family regarding lab results while reviewing another patient with a different trainee), we may be able to meet family needs better, without losing time efficiencies. Other communication avenues that spare time-loss could include innovative digital adaptations that allow families to see wait times or access their own lab results [29]. These findings underscore the importance of caregiver involvement in their child's care and adequate communication by ED staff as critical driving factors in improving emotional and communication needs and satisfaction with care for caregivers in the ED. These key elements may also impact the provision of confident and potentially better post-discharge care at home.

Caregivers in our study had significantly high levels of anxiety during their child's ED visit, with a mean score of almost 38. Such high levels have previously been demonstrated in other pediatric ED studies as well [24, 30], ranging from 32 to 42. In comparison, baseline state STAI scores for Indian soldiers is 39, [31] and for German soldiers is 32 [32]. Further, our study demonstrated that for every 1 unit of increase in anxiety (STAI score), there was a 3% increase in the odds ratio for both emotional needs being unmet and increased caregiver discomfort in caring for their child's illness at home. It has been previously shown that caregiver anxiety levels are higher with greater acuity of their child's illness; [2] our study found higher acuity to be associated with caregiver emotional needs more likely to be met. It is possible that health care providers spend more time with the sickest families, particularly in the first few critical hours of ED admission, providing more frequent updates and answering questions and concerns. HCPs likely recognize the importance of timely and thorough communication, yet perhaps lack the time or staffing to maintain this standard for families with lower acuity presentations. Interestingly, high anxiety and reports of caregiver dissatisfaction before physician assessment suggests that the waiting room experience may play an important role in how families will perceive the remainder of their clinical experience and optimization of waiting room processes could lead to overall improved experiences [13, 24].

Prior literature suggests that one third of caregivers seeking care for their children in the ED demonstrate low health literacy and this has been identified as an independent predictor of higher ED use and presentation to the ED for non-urgent conditions [26, 33]. Our study demonstrated a similarly low health literacy in 28.5% of participants, and further showed that for every 1 unit of increase in health literacy score, there was a 9% increase in odds ratio for caregiver emotional needs being met. This suggests that when explaining and discussing medical matters with families, we must be aware that 1/3 of families will have reduced health literacy and our communication needs to be adapted to this. Further, these families are at higher risk for emotional needs not being met, very possibly related to not understanding what is being shared, potentially leading to discomfort in caring for their child at home with more return visits to the ED. Clinicians may consider utilizing "universal precautions" as an approach to improve the care of patients with low health literacy [26]. This approach includes improved

communication from all members of the health care team, provision of written education materials to support understanding, utilizing a teach back method and employing interactive communication processes such as oral, written, graphical or online resources [26].

Our findings have reaffirmed the complex relationship between emotional needs and communication, both key concepts in PFCC. It has been well established that when pediatric patients and their caregivers are understood and honored in the context of their family, culture, values, and goals, this results in better health care, safety and patient satisfaction [34, 35]. Our study demonstrated that involving caregivers in their child's care increased caregiver emotional needs being met and increased caregiver comfort in caring for their child's illness at time of discharge. Involving caregivers recognizes and builds on the strength of individual children and families, empowering them to discover their own strength and build confidence [11, 34, 35]. These positive health care experiences within this partnership enhance caregivers' confidence in their roles [11], which can lead to increasing comfort in caring for their child's illness at discharge. Communication in the form of providing satisfactory updates and answering caregiver questions and concerns were key variables in both meeting caregiver emotional needs and increasing caregiver comfort in caring for their child's illness at time of discharge. These findings reaffirm the notion that communication is the cornerstone of excellent PFCC care, creating a safe space for shared decision-making [3, 34].

This study supports the existing literature that caregivers of children present to the ED with significant anxiety [36, 37]. This anxiety can be greatly impacted by PFCC interventions to reduce stress, anxiety, and depression, increase satisfaction with care, and improve relationships with health care providers [38]. Recognizing that increased caregiver anxiety decreases emotional needs being met creates a window of opportunity to address these stressors as soon as the patient and family caregivers enter the ED [24]. Providing health information to caregivers that considers health literacy, is delivered by numerous members of the health care team, and is available in a range of formats (e.g., written, oral, graphical, linguistic diversity) is aligned with the principles of PFCC [11]. This may include standard discharge instructions that are customized to reflect family preferences, in addition to printed or online resources, as they reassume complete care of their child at discharge [34].

## Conclusion

In this study, we found that for caregivers presenting to the pediatric ED, approximately 30% have unmet emotional needs, over 15% have unmet communication needs, and 15% of caregivers felt inadequately involved in the care of their child, challenging us as HCPs to do better. These unmet needs negatively influence caregivers' satisfaction with care, but even more importantly, their ability to comfortably care for their child's illness at home. We found that involving caregivers in their child's care and adequate communication were key driving forces in meeting the emotional and communication needs of families. For this reason, immediate changes such as improved communication via more frequent updates from varied members of the health care team, adequately answering of questions and concerns, and involving caregivers directly in the care of their child can improve the ED experience. More complex systemic interventions (e.g., addressing of national ED overcrowding and staffing challenges) and public health initiatives to address caregiver health literacy and anxiety surrounding ED visits will also make a positive impact on family ED experiences; these will require concerted national efforts to rectify.

## Limitations

Surveys were administered in only English or French, limiting the perspectives of others with diverse linguistic and cultural backgrounds, and the generalizability of our findings. Despite

our efforts to ensure robust survey tools, self-reporting may introduce recall or response biases. We attempted to mitigate this through careful selection of research questions, the administration of two surveys (one during the visit and a second one in follow-up), choosing an appropriate data collection method for the population (iPad or email), and collecting the data prospectively. The research assistants administered the surveys during daytime and evening hours, as a convenience sample, perhaps not capturing the unique needs of caregivers during the overnight period. This study focuses on caregiver experiences only, not capturing the emotional needs of the full family (e.g., child, other caregivers). The survey also excluded caregivers whose child was medically unstable, therefore we may not have captured the emotional needs of caregivers with the sickest children. As such, this study may not fully represent the diversity of all families visiting pediatric EDs.

## Supporting information

**S1 Data.**
(PDF)

**S1 Table. ED visit details.**
(DOCX)

**S2 Table. Univariable logistic regression model for likelihood of emotional needs being met.**
(DOCX)

**S3 Table. Univariable logistic regression model for caregiver comfort in caring for their child's illness at home.**
(DOCX)

## Acknowledgments

To the families for their voluntary participation in our study, we offer our sincere gratitude. In addition to the above-named authors, the PERC Family Needs Study team includes Ashley Jones, Dr. Ran Goldman, Dr. Darcy Beer, Rachel Keijzer, Tannis Erickson, Kamary Coriolano, Dr. Amy Plint, Candice McGahern, Tyrus Crawford, Dr. Laura Weingarten, Dr. April Kam, Wenli Xie, Bethany Lerman, Marie-Christine Auclair, Dr. Janet Curran, Dr. Katie Gardner, Yvonne Suranyi, Christine Westerlund, Dr. Lisa Hartling, Patricia Candelaria, and Kurt Schreiner. Dr. Samina Ali (sali@ualberta.ca) is the lead author for this group. We thank them for their contributions to this study.

## Author Contributions

**Conceptualization:** Samina Ali, Claudia Maki, Keon Ma, Maryna Yaskina, Antonia Stang, Naveen Poonai, Manasi Rajagopal, Julie Leung, Bruce Wright, Shannon D. Scott.

**Data curation:** Samina Ali, Asa Rahimi, Helen Wong, Antonia Stang, Tania Principi, Naveen Poonai, Serge Gouin, Sylvia Froese R. N., Paul Clerc, Redjana Carciumaru, Waleed Alqurashi, Manasi Rajagopal.

**Formal analysis:** Samina Ali, Claudia Maki, Maryna Yaskina.

**Funding acquisition:** Samina Ali, Shannon D. Scott.

**Investigation:** Samina Ali, Shannon D. Scott.

**Methodology:** Samina Ali, Keon Ma, Maryna Yaskina, Elise Kammerer, Julie Leung, Shannon D. Scott.

**Project administration:** Samina Ali.

**Resources:** Bruce Wright.

**Supervision:** Samina Ali, Shannon D. Scott.

**Visualization:** Elise Kammerer.

**Writing – original draft:** Samina Ali, Claudia Maki.

**Writing – review & editing:** Asa Rahimi, Keon Ma, Maryna Yaskina, Helen Wong, Antonia Stang, Tania Principi, Naveen Poonai, Serge Gouin, Sylvia Froese R. N., Paul Clerc, Redjana Carciumaru, Waleed Alqurashi, Manasi Rajagopal, Elise Kammerer, Julie Leung, Bruce Wright, Shannon D. Scott.

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
