## [Decision Letter · Decision Letter 0]

31 Aug 2023

PONE-D-23-17555Family caregivers’ emotional and communication needs in Canadian pediatric emergency departmentsPLOS ONE

Dear Dr. Samina Ali,

Thank you for submitting your manuscript to PLOS ONE. After careful consideration, we feel that it has merit but does not fully meet PLOS ONE’s publication criteria as it currently stands. Therefore, we invite you to submit a revised version of the manuscript that addresses the points raised during the review process.Present your study limitation at the last paragraph of discussionrecommendation for future researchPlease submit your revised manuscript by Oct 15 2023 11:59PM. If you will need more time than this to complete your revisions, please reply to this message or contact the journal office at plosone@plos.org. Please include the following items when submitting your revised manuscript:A rebuttal letter that responds to each point raised by the academic editor and reviewer(s). You should upload this letter as a separate file labeled 'Response to Reviewers'.A marked-up copy of your manuscript that highlights changes made to the original version. You should upload this as a separate file labeled 'Revised Manuscript with Track Changes'.An unmarked version of your revised paper without tracked changes. You should upload this as a separate file labeled 'Manuscript'.If applicable, we recommend that you deposit your laboratory protocols in protocols.io to enhance the reproducibility of your results. Protocols.io assigns your protocol its own identifier (DOI) so that it can be cited independently in the future. For instructions see: https://journals.plos.org/plosone/s/submission-guidelines#loc-laboratory-protocols. Additionally, PLOS ONE offers an option for publishing peer-reviewed Lab Protocol articles, which describe protocols hosted on protocols.io. Read more information on sharing protocols at https://plos.org/protocols?utm_medium=editorial-email&utm_source=authorletters&utm_campaign=protocols.

We look forward to receiving your revised manuscript.

Kind regards,

Fadwa Alhalaiqa

Academic Editor

PLOS ONE

Journal Requirements:

"This study was supported by the Women and Children’s Health Research Institute (WCHRI) and the Clinical/Community Research Integration Support Program (CRISP) grant, secured by Dr. Shannon Scott and Dr. Samina Ali (2018-2022)."

5. One of the noted authors is a group or consortium the Pediatric Emergency Research Canada Family Needs Study Group. In addition to naming the author group, please list the individual authors and affiliations within this group in the acknowledgments section of your manuscript. Please also indicate clearly a lead author for this group along with a contact email address.

6. We are unable to open your Supporting Information file Appendix 1 Family Needs Survey 6jun2023.pdf. Please kindly revise as necessary and re-upload.

Additional Editor Comments:

Dear Authors

Your team effort is really appreciated in writing this paper. However, you need to meet the reviewers feedback

Reviewers' comments:

Reviewer's Responses to Questions

**Comments to the Author**

1. Is the manuscript technically sound, and do the data support the conclusions?

Reviewer #1: Yes

Reviewer #2: Yes

2. Has the statistical analysis been performed appropriately and rigorously? 

Reviewer #1: Yes

Reviewer #2: Yes

3. Have the authors made all data underlying the findings in their manuscript fully available?

Reviewer #1: Yes

Reviewer #2: Yes

4. Is the manuscript presented in an intelligible fashion and written in standard English?

Reviewer #1: Yes

Reviewer #2: Yes

5. Review Comments to the Author

Reviewer #1: Thank you very much for inviting me to read this good manuscript that highlights the importance of doctors and nurses in emergency units and the impact this has on child care providers. I will allow myself to make the following observations to the manuscript to raise its academic quality:

Introduction: OK

Methods:

The described methodology of the manuscript presents several strengths and noteworthy considerations. The study's strengths include its comprehensive approach to data collection and analysis. The sub-study employs a descriptive cross-sectional survey coupled with medical record review, providing a multi-faceted view of caregivers' experiences in pediatric emergency departments (EDs). The utilization of a convenience sample from ten Canadian pediatric EDs enhances the study's external validity, making the findings more generalizable. The rigorous development and validation process of the survey tools, involving expert panel input, pre-testing, pilot testing, and sensibility testing, contribute to the robustness of the measures used to assess caregiver needs and experiences.

However, there are also potential challenges and limitations to consider. The use of a convenience sample might introduce selection bias, as participants may not fully represent the diversity of all families visiting pediatric EDs. Despite efforts to ensure robust survey tools, self-reporting may introduce recall or response biases. I recommend that the authors describe in more detail how they did to reduce the risk of bias in the study.

Results: OK

Discussion: OK

Reviewer #2: General: A very well prepared manuscript and clearly laid out methodology.

Abstract; Objectives: Note that the objectives in the abstract are not in alignment with the objectives in the article.

Abstract; Objectives: Would you say that the findings presented in this article related the findings to the demographic characteristics? Perhaps rework to be in line with the objective presented in the article.

Introduction: Consider including a brief description of the term "caregiver satisfaction". This is obviously similar to but not the same as patient satisfaction which is not applicable to this study.

The term "team" is utilised quite frequently. And in some instances is referred to as ED team. It would be helpful to the reader to indicate early on who is being referred to when the term "team" is used. In some EDs there may be designated nurse who would be a "constant" throughout the caregiver's time in ED.

First and only time the PICU acronym is used. Recommend writing in full. Just a query if this should be ED setting?

Page 5; Line 7: What is meant by "Characterizing the emotional and communication needs...."?

Page 5; Line 13: The term "satisfaction" is presented alone. It needs qualification. Satisfaction with what?

Page 5; Line 14: The term "visit characteristics" also needs elaborating upon

Page 7; Line 13: Was the word "permuted" intentional? If yes, maybe change from "missing responses" to "missed responses".

Page 8; Line 16: Clarify: satisfaction with what?

Page 18; Line 15: Is it to improve the overall patient experience or the overall caregiver experience? - the focus of this study being on the caregiver.

Page 18; Line 20: Kindly provide some literature to show whether existing data supports/contradicts these findings.

Page 19; Line 1-4: As above, kindly show how existing research supports/contradicts these findings.

Page 19; Line 20: Kindly elaborate on/qualify the term satisfaction, i.e. satisfaction with what?

Page 20; Line 7-9: Kindly note that anxiety levels and emotional needs are not necessarily the same thing. One may have one's emotional needs met, but anxiety is still present. Rather find another study that refers to emotional needs. However, it is an interesting outcome.

Page 22; Line 5: Increase satisfaction with what?

It is recommended that the conclusion also includes the many positive outcomes of the findings. For instance, the majority of caregivers had their emotional needs met and the majority of caregivers had their communication needs met. And this is across Canada's pediatric EDs. This is a highly commendable outcome.

Page 22; Conclusion, Line 4: Does the word "influence" refer to influence for the better or worse? The apostrophe for caregivers should be after the "s" as caregivers'. And satisfaction with what?

Page 23; Conclusion; Last sentence: The phrase "make an impact" - kindly clarify: Make an impact on what?

6. PLOS authors have the option to publish the peer review history of their article (what does this mean?). If published, this will include your full peer review and any attached files.

Reviewer #1: No

Reviewer #2: No

---

## [Author Response · Author response to Decision Letter 0]

28 Oct 2023

Please see uploaded 'Response to Reviewer' table

---

## [Editor Report · Decision Letter 1]

5 Nov 2023

Family caregivers’ emotional and communication needs in Canadian pediatric emergency departments

PONE-D-23-17555R1

Dear Dr. Samina Ali,

We’re pleased to inform you that your manuscript has been judged scientifically suitable for publication and will be formally accepted for publication once it meets all outstanding technical requirements.

Kind regards,

Fadwa Alhalaiqa

Academic Editor

PLOS ONE
---

## [Editor Report · Acceptance letter]

9 Nov 2023

PONE-D-23-17555R1 

Family caregivers’ emotional and communication needs in Canadian pediatric emergency departments 

Dear Dr. Ali:

I'm pleased to inform you that your manuscript has been deemed suitable for publication in PLOS ONE. Congratulations! Your manuscript is now with our production department. 

Kind regards, 

on behalf of

Pro Fadwa Alhalaiqa 

Academic Editor

PLOS ONE